# The Impact of Urban Allotment Gardens on Physical and Mental Health in Norway

**DOI:** 10.3390/ijerph21060720

**Published:** 2024-05-31

**Authors:** Mari Johnsrud, Ursula Småland Goth, Hilde Skjerve

**Affiliations:** 1Department of Health and Exercise, School of Health Sciences, Kristiania University College, 0107 Oslo, Norway; marijohnsrud@hotmail.com; 2NLA University College, Campus Oslo, 0103 Oslo, Norway; ursula.goth@nla.no

**Keywords:** allotment house, allotment garden, quality of life, physical health, mental health, perceived health

## Abstract

In Norway, many communities offer low-rent plots of land on which city dwellers can build summer cottages and grow crops. These allotment gardens serve as recreational escapes from urban life. However, little is known about the impact these gardens have on their members. This study attempts to shed light on today’s impact of allotment gardens in a public health setting in Norway. The study was based on 17 semi-structured interviews and 2 in-depth interviews with allotment house owners. Informants were mainly recruited by “snowball sampling”. Our data show that maintaining an allotment promotes exercise and provides a positive impact on self-perceived well-being and physical health through outdoor activities. Spending time in the garden contributes to new knowledge and experiences. Owning an allotment garden has provided new knowledge, new skills, new hobbies, and thereby an improvement in lifestyle. The allotment garden has a strong impact on perceived health, well-being, and sense of coherence (SOC) for the individuals. It promotes outdoor activities along with social interaction and can prevent feelings of loneliness and isolation.

## 1. Introduction

Allotment gardens often include simple cabins, usually built out of recycled materials on a small plot of land. The garden plots are given to local community members to serve as getaways from city life with opportunities to grow one’s own vegetables and live closer to nature. The first allotment gardens were established in Norway in 1907. At the time, many Norwegians, especially those living in big cities like Oslo, faced difficult social conditions. Allotment gardens were therefore established to improve the lives of city dwellers. Today in Oslo alone, there are nine separate allotment gardens with 1097 houses, and as of 31 May 2022, there were 7672 people on a waiting list to obtain one [1,2]. The gardens are open for use between 1 April and 1 October. During the fall and winter, the water supply and sewage are cut off to avoid freezing.

There is a growing consensus that the places where people live and the various social processes, relationships, and psychosocial concepts are associated with healthy communities, and that neighborhoods contribute to health [3]. Only recently have studies begun to shed light on the allotment gardens’ positive impact on experienced health, lifestyle, and activity levels [4] due to the higher intake of fruits and vegetables [5,6,7] and the reduction in lifestyle diseases [4,5,8,9].

In recent years, the number of people with mental health challenges has increased [10]. In the Coyle and Dugan study, there was an observed connection between loneliness and a higher risk of having some form of mental illness [11]. In one survey, 62% of those persons who were experiencing loneliness answered that they were dissatisfied with their lives ([12] p. 49). Elderly people (65+) are more likely to be affected by loneliness [13,14].

Fortunately, several studies have found that activities such as gardening helped to reduce the incidence of social isolation and exclusion [5,6,15]. Findings from Martens et al. showed that the allotment garden had a positive impact on the quality of life for older gardeners [16]. Gardening has been shown to prevent the development of poor self-rated health [15,17] and improve feelings of mental well-being [5,6]. People with cognitive disorders can benefit from gardening activities [5,6] as they can have a therapeutic impact on health both mentally and physically [18]. One study found that gardening had a positive effect on mental health regardless of the person’s state of health [17]. Martens indicated that allotment gardens should be used as part of a low-threshold preventive health measure [16]. Being part of a social environment can have a positive impact on stress management and, under the right circumstances, lead to increased internal motivation and better self-regulation ([19] p. 16).

This study aims to shed light on the impact of allotment gardens today in a public health setting in Norway. By doing so, the study aims to identify factors that improve self-perceived health, well-being, and the impacts on inhabitants’ everyday life.

## 2. Materials and Methods

This study was conducted in 2022 and is based on 17 semi-structured interviews and 2 in-depth interviews with allotment house dwellers. Four of the informants were between 31 and 55 years of age and the remaining fifteen informants were in the 56–85 age group.

### 2.1. Interview Guide and Recruitment

A semi-structured interview based on mapping, lifestyle in the allotment garden, and the perceived importance of the allotment garden (Appendix A) was developed in collaboration with allotment house owners. All informants were over 18 years of age and had owned an allotment for at least six months. Informants were recruited via “snowball sampling”, whereby study subjects are invited to recruit new subjects in an ever-growing, self-enlarging sample ([20] pp. 21, 77) and via an advertisement in one garden’s Facebook group. The inclusion of 17 interviews during the COVID-19 restrictions showed data saturation. During the process of writing the article after the restrictions were lifted, we wanted to ensure data saturation and included two more interviews in the study. ([20] pp. 78–79). The in-depth interviews with allotment owners took place at their compound and lasted between 65 and 90 min.

### 2.2. Interview Method and Analysis of the Data

The initial study took place during the COVID-19 restrictions. As a consequence, only 13 interviews could be performed in person. Three interviews were conducted over the telephone, and one took place through a Zoom video meeting. The telephone interviews lasted from 25 to 40 min while the physical semi-structured interviews lasted a duration of 30–90 min. The telephone informants answered the questions more comprehensively and directly than the physical interviewees did.

After the COVID-19 restrictions were lifted and the first analysis was performed, data from two additional in-depth interviews were collected at the allotment compound. Those interviews lasted between 60 and 90 min.

Before an interview took place, the informants were told about the purpose of the study, the study design, and our intention to publish the results ([20] Ch. 5). The interviews were performed by M.J. and U.S.G. After each interview, the researcher showed the informant their notes and quotes, which the informant verified [21]. Only verified notes and quotes were included in the data. Immediately after the interview, the interviewers reflected on what was said This allowed us to register more reflections. The reflection took place immediately after the interview and reduced any recall bias ([20] p. 118).

The analysis was performed in collaboration with the authors. During the analyses, a template for deduction was used [22]. This stepwise analysis was an aid to coding and dividing the data into thematic groups [23]. The analysis divided our data into three thematic groups: (1) well-being and physical activity; (2) social activity/sense of coherence (SOC) and routine; and (3) place for the family. To reduce potential bias and assess the data’s consistency, the authors validated the results, which were then summarized to give an overview. The analysis was seen as a process of examining and interpreting data to elicit meaning, gain understanding, and develop empirical knowledge. Empirical data were collected inductively and after that coded and grouped. Pre-existing theories found within the previous analysis were compared with data from the following interview.

### 2.3. Weaknesses of the Study That May Have Affected the Result

By using “Snowball recruiting” instead of collecting a random sample, representativeness could not be guaranteed. Therefore, a sampling bias might have occurred.

Snowball recruiting also had an impact on gender distribution. Even though we encouraged equal participation of both genders, only five men participated in the study.

We used no recording devices during the interviews to get the informants to speak freely, implicating no independent means to check the accuracy of the notes.

### 2.4. Ethics

The study followed the ethical guidelines of Norwegian Centre for Research Data that state that participation in research should be based on information, consent, anonymity and confidentiality. An informed consent form was given to each informant prior to starting the interview. Data were collected anonymously so no identifying information was collected and interviewers’ responses were recorded through handwritten notes. This way, the study did not require ethical approval [24].

## 3. Results

### 3.1. Well-Being and Physical Activity

“*Since I struggle with musculoskeletal disorders, I have days where I must pay extra attention and listen to the body when it needs rest. But gardening helps me to maintain normal function*.”(Informant 1)

“*I used to look at weeding as a chore, whereas now it’s a bit like meditation*.”(Informant 10)

“*Being at my allotment house makes me feel young and active. I have a place to be with my family, socialize with my friends, and at the same time be physically active*.”(Informant 18)

Our findings show that keeping an allotment is a great form of exercise. Most free time goes to gardening. Younger members had less time because of childcare and job requirements. Informants older than 56 spent more time growing plants, weeding, and carrying out maintenance work. Weather and time of year were the two factors affecting the amount of activity in the garden. During the winter, only short visits were made to the garden.

### 3.2. Life-Style: Experience and Knowledge

“*My partner and I have learned a lot about cultivation and got green thumb after acquiring an allotment garden. We have become better at utilizing the raw materials we have in the garden*.”(Informant 3)

“*Through active gardening, I have experienced both physical and mental mastery because I get to do new things*.”(Informant 1)

“*I have learned to be more outdoors and become more aware of nature*.”(Informant 4)

Spending time in an allotment garden contributes to new knowledge and experiences, and this contribution had an impact on lifestyle. As Table 1 indicates, all informants stated that they had gained new knowledge after acquiring an allotment garden. Fruit and vegetable cultivation, beekeeping, and tree pruning were some of the activities the informants mentioned. They affirmed that they gained knowledge through their own experience, through friends, family, or other gardeners, and that they experienced mastery through various activities.

Table 1, which is based on the interviews, illustrates that informants over 55 took more time to participate in social activities or voluntary work. Social interaction was more important for allotment owners between 56 and 85 years, and all participants expressed the importance of socializing with family and friends. Finally, all our informants gained interest and knowledge in growing vegetables and fruits and by maintaining the allotment house.

### 3.3. Social Activity and Sense of Coherence

“*I think that having access to an allotment garden is the biggest factor that protects against loneliness, depression, isolation, and diseases. And you get a sense of belonging. It also helps to even out socio-cultural differences*.”(Informant 4)

“*The social life in the allotment garden has meant a lot for my health and well-being, especially during COVID-19. Had it not been for the allotment garden, I probably would have climbed the walls*.”(Informant 5)

As the quotes reveal, social interactions with family, friends, and other gardeners were and are an important part of life in the allotment community. For most of our informants the garden can counter loneliness. Particularly, vulnerable people and those over 55 will find SOC through social interaction with others (Table 1). Our data also showed that the social environment was especially important among those who spent every day in the garden during the growing season. This was not the case for allotment owners under 55. For them, work, childcare, and household activities took up most of their time; therefore, they could not spend every day in the garden.

Social events are a central part of the allotment garden. Most informants were positive about attending events. Only 2 of the 19 informants stated that they never or rarely participated in social events.

The feeling of belonging and social interaction is central to one’s well-being. During the outbreak of COVID-19 in 2020, two of the informants mentioned that the garden was a life saver for their mental, physical, and social health.

### 3.4. Daily Routine and Physical Activity

“*I feel a responsibility to manage the existing natural basis that is in the garden in the most sustainable way possible*.”(Informant 6)

“*I feel a responsibility that my daughter should have a nice and green place to grow up*.”(Informant 4)

As our data show, many of the informants’ lives centered around he allotment garden. Many informants also mentioned the garden’s positive impact on both physical and mental health. For instance, informants that recently experienced disruptive situations like retirement or divorce found allotment activities to be a stabilizing factor in their lives.

Our data also indicate that, by keeping an allotment garden, which requires a lot of hard work and advanced planning, our informants had the feeling of SOC and mastery when seeing a project through from start to finish.

### 3.5. Place for the Family

“*I cannot afford a garden and a house. It would have been a completely different upbringing for my daughter if we had not had an allotment garden. There’s the freedom, the belonging, and things she has learned from being here. She would have had a much poorer upbringing without the allotment garden. This also applies to many of her friends from school and the neighborhood*.”(Informant 4)

“*The allotment house gives me the opportunity to interact with my family and see several members at the same time. It looks like the garden has been a place where we meet and have time for each other*.”(Informant 17)

“*The allotment garden is a great place for all of us. Here, we can also teach our children where the food comes from*.”(Informant 2)

## 4. Discussion

Communities like those we find at urban allotment gardens are more likely to experience better self-reported health and well-being [3]. Allotment gardens have been in existence since 1907, yet we still have little knowledge about their benefits for the people who tend them [3,25].

### 4.1. Well-Being and Physical Activity

Salutogenesis predicts that organized physical activity can be a health resource [13,14]. Previous studies have confirmed that an allotment garden lowers the threshold for being outdoors and socializing with other people [6,8,18]. We found similar results in our study, indicating that allotment gardens have a significant impact on well-being and physical health and thereby on the individual’s mental health.

### 4.2. Knowledge

Living in an allotment garden has provided new knowledge, new skills, new hobbies, and therefore a change in lifestyle. The experience of learning and mastering has a beneficial effect on self-development and self-esteem in humans [19]. Genter et al. found that allotment gardens could facilitate self-development and a sense of mastery [18]. In our study, several informants described their involvement with activities such as beekeeping, gardening, organizing social events, or coffee meetings. As other studies [8] show, the feeling of belonging and SOC is essential for one’s well-being, and we saw how an allotment garden fulfilled at least some of those needs.

### 4.3. Mental Health

Green spaces can play an important role in health promotion ([26] Ch. 18). During the COVID-19 pandemic, there was considerable research about the role of private gardens and digital nature, which demonstrated that natural environments have the potential to buffer the impact of stressful events [27]. Lachowycz and Jones suggest that there are psychological benefits that can be derived from contact with nature such as stress reduction and positive emotions [28]. Seen from a salutogenic perspective, physical activity in an allotment garden has been shown to be a meaningful, comprehensible, and manageable way for older people to maintain their health [26] and that gardening might help reduce stress [6,18,29]. In addition, our findings show that gardening can be recommended for elderly people, ethnical minorities, and people with long-term health challenges by preventing social exclusion. Our finding was also seen in a study by Hajek and Köning [15,30].

### 4.4. Social Interactions, Room for Family and Friends, and the Perceived Significance of the Allotment Garden

SOC is a construct that refers to the extent to which one sees one’s world as comprehensive, manageable, and meaningful [27]. In our study, we saw that the nature of the links between coherence and adaption reinforced each other.

Social interaction is an essential part of everyday life in an allotment garden, and especially important for vulnerable groups such as, the elderly, migrants, and single families, for example [30]. Gardeners contribute to and benefit the local environment [4], which increases social interaction [25]. Several informants pointed out their participation in various groups and events, and our study results show that carrying out gardening was important for their well-being.

The allotment garden’s role in residents’ well-being must not be underestimated [31]. It is therefore natural to understand that the loss of an allotment garden will affect a person’s health.

## 5. Conclusions

Clearly, living in a garden house boosts one’s health and social life and could be an important facility for vulnerable groups in the population. As our data show, allotment gardens are experienced as an arena for the inclusion of vulnerable groups such as the elderly, migrants, or single parents for various reasons. Therefore, our study concludes that the impact of public allotment gardens is characterized by a social and physically active lifestyle. Age, life situation, and interests determine how the residents utilize their garden. A garden has a significant impact on the lives of the owners, family members, friends, and the public visiting the compound. A garden has a strong impact on perceived health, well-being, and SOC for the individual, leading to increased outdoor activities and social interaction. While the findings suggest potential benefits, more comprehensive research is needed to confirm these outcomes.

### Further Research

Further research is needed. Until recently, there have been few scientific studies on allotment gardens. This study used a qualitative approach and covered only one of Norway’s allotment gardens. Follow-up studies could include other gardens within Norway, Denmark, or Austria where allotment gardens are common, and include a balance of genders, ages, and educational levels.

## Figures and Tables

**Table 1 ijerph-21-00720-t001:** Usage of the allotment house and social interaction.

	31–55 Years (Percentages)	56–85 Years (Percentages)
Stay during season		
Full time	1 (25%)	11 (73%)
Part time	3 (75%)	4 (27%)
Garden (social) events and voluntary work		
Participated	1 (25%)	8 (54%)
Participated if required	3 (75%)	5 (33%)
Did not participate	0	2 (13%)
Importance of SOC/social interaction		
Participated in various activity groups and events	4 (100%)	13 (87%)
Did not participate in activity groups or events	0	2 (13%)
Was social with family, friends	4 (100%)	15 (100%)
Was not social with family, friends	0	0
Gained interest and knowledge (maintenance and gardening)		
Learned something new	4	15

## Data Availability

The original contributions presented in the study are included in the article, further inquiries can be directed to the corresponding author.

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
