# Peer review of "The Impact of Urban Allotment Gardens on Physical and Mental Health in Norway"

_ijerph, 2024, doi:10.3390/ijerph21060720_

Round 1

Reviewer 1 Report

Comments and Suggestions for Authors

The article, which discusses urban allotment gardens in Norway and their effects on public health, emphasizes the benefits these gardens offer in promoting physical health, social interaction, and improved mental well-being.  It is designed as a qualitative method with semi-structured and in-depth interviews. This is a very interesting and important study contributing to the public health of communities, especially for older communities. Please see my minor comments below.

Minor comments:

Page 1 Line 2- please consider adding “Physical and Mental Health” for example  "The Impact of Urban Allotment Gardens on Physical and Mental Health in Norway"

Page 1 Line 21- please consider adding the following keywords: Perceived health, Physical Health, Mental health, Norway

Page 1 line 29- Please correct the formatting of the following sentence “improve the lives of city dwellers”

Page 1 Line 43- Please correct the formatting of the following sentence “persons who were experiencing loneliness”

Page 2 Line 61- There is a consistency between the numbers of semi-structured interviews in the abstract and method section. (17 semi-structured interviews in the abstract; 19 semi-structured in the method section).

Page 2 Line 70-71- Provide a clearer explanation of how data saturation was determined.

Page 2 Line 71- Please justify for additional two semi-structured and in-depth interviews after the data saturation has been reached.

Page 3 Lines 106-108- Please consider mentioning the specific guidelines followed, such as consent, confidentiality, and anonymity.

Page 6 Line 242-243- The following conclusion might be overly optimistic based on the qualitative data “As a result, allotment gardens can enhance well-being and quality of life." Please state that “while the findings suggest potential benefits, more comprehensive research is needed to confirm these outcomes.”

Page 6 Lines 266-267- please consider adding websites for ref 1 and ref 2.  

Best regards, 

Author Response

Thank you so much for your comments. All your comments are now included in the article

Reviewer 2 Report

Comments and Suggestions for Authors

Thank you very much for this interesting topic. I was really looking forward to reading this paper.

Firstly, I would like to offer a little constructive criticism, and then raise a more general point:

Introduction

„After 15 interviews, the data showed saturation [20] p. 78-79]. Thereafter, two semi-structured interviews and two in-depth interviews were carried out [20 Ch. 5].“

This sentence is confusing - first, it is said that 19 semi-structred interviews were conducted and 2 in-depth interviews. How does this fit?

Material and Methods

Why were the two methods of semi-strcutured and in-depth interviews used and not any others? An explanation would be useful

Why was there a change in the method of data generation, and not always telephone or always Zoom? Please explain.

"Immediately after the interview, the interviewers held a reflection on what was said This allowed us to register more reflections. Time for reflections immediately after the interview reduced any recall bias [20, p. 118]."

The reflections were no longer verified by the interviewees, is that correct? Why is that? Please explain.

Which method was used for the coding process to analyse the data? Please provide more information on how the dataset was analysed and the results generated.

Results

Quotes and results do not match; e.g. in chapter 3.1., but also in the others. The results are not discussed in detail, but are partly listed in one sentence each.

Table 1: Please also indicate percentages (there were only 4 people in the younger group - i.e. if 4 people gave a certain answer, this corresponds to 100& of this age group).

Discussion

As to the results, the discussion remains on a superficial level. More "meat to the bone" would be great

General remark

Unfortunately, the results/discussion sections of the paper mainly repeat aspects that are already known (general positive effects of gardening and spending time in a garden) - I am missing the special features of allotment gardens in that context. What characterises an allotment garden in terms of positive effects? What makes it special? And perhaps there are also findings in the results that show effects in more detail? I am a great fan of such kind of methods and the topic of allotment gardens in general, but I miss a certain detailed level regarding the results and discussion/interpretation of the results and the singularity of allotment gardens.

Author Response

Thank you so much for your comments. The most of your comments are now included in the article. We have also changed the title on the article from public health to physical and mental health in Norway
